# Individual pupil size changes as a robust indicator of cognitive familiarity differences

**Léon Franzen**[1,2,3]⊛*, **Amanda Cabugao**[1]⊛, **Bianca Grohmann**[2], **Karine Elalouf**[1], **Aaron P. Johnson**[1,4]

**1** Department of Psychology, Concordia University, Montréal, Quebec, Canada, **2** Department of Marketing, John Molson School of Business, Concordia University, Montréal, Quebec, Canada, **3** Department of Psychology, University of Lübeck, Lübeck, Schleswig-Holstein, Germany, **4** Vision Health Research Network, Montréal, Quebec, Canada

⊛ These authors contributed equally to this work.
* leon.franzen@mail.com

**Data Availability Statement:** All data files supporting the results are available from the project's Open Science Framework repository (https://osf.io/3w5s6/). These files provide access to single-trial pupil size data and participant

## Abstract

Cognitive psychology has a long history of using physiological measures, such as pupillometry. However, their susceptibility to confounds introduced by stimulus properties, such as color and luminance, has limited their application. Pupil size measurements, in particular, require sophisticated experimental designs to dissociate relatively small changes in pupil diameter due to cognitive responses from larger ones elicited by changes in stimulus properties or the experimental environment. Here, building on previous research, we present a pupillometry paradigm that adapts the pupil to stimulus properties during the baseline period without revealing stimulus meaning or context by using a pixel-scrambled image mask around an intact image. We demonstrate its robustness in the context of pupillary responses to branded product familiarity. Results show larger average and peak pupil dilation for passively viewed familiar product images and an extended later temporal component representing differences in familiarity across participants (starting around 1400 ms post-stimulus onset). These amplitude differences are present for almost all participants at the single-participant level, and vary somewhat by product category. However, amplitude differences were absent during the baseline period. These findings demonstrate that involuntary pupil size measurements combined with the presented paradigm are successful in dissociating cognitive effects of familiarity from physical stimulus confounds.

## Introduction

When completing shopping tasks, consumers frequently seek to identify familiar branded products. In light of crowded shelves and product proliferation, consumers may find it difficult to report their level of brand familiarity with a high level of confidence. This is just one of many situations that can give rise to inaccurate cognitive judgements of familiarity in consumers' self-reports. As this example highlights, a key problem with self-reports is their potential inaccuracy and susceptibility to cognitive biases—even though self-reports are convenient for researchers [1, 2]. In an attempt to preclude self-report biases, consumer psychology has only

averages of those data. Data underlying the initial image validation part of the study—averaged by image—are equally available from the OSF repository.

**Funding:** This work was funded by a Horizon Postdoctoral Fellowship awarded to LF by Concordia University (https://www.concordia.ca), and by a Fonds de Recherche du Québec - Société et Culture (http://www.frqsc.gouv.qc.ca/en/) team grant to BG and AJ (grant 196369). The funders had no role in study design, data collection and analysis, decision to publish, or preparation of the manuscript.

**Competing interests:** The authors declare no competing interests.

recently started to adopt objective physiological measures, such as pupillometry, eye-tracking and electroencephalography (EEG), which capture implicit processes that are difficult to articulate or express accurately otherwise [1, 3].

Pupillometry is an implicit, physiological measure that traces changes in pupil size that predominantly occur due to the pupillary light reflex adjusting the pupil to incoming light [4, 5]. The incoming light depends on stimulus properties, such as luminance and color (reviewed in [6]). The pupil takes about 500 ms to respond to a stimulus [7] due to the delay in the iris muscle contraction [8], and generally peaks around 1000 ms post-stimulus onset [9]. Changes in pupil size have been associated with activation of the sympathetic (dilation/increase) and parasympathetic (constriction/decrease) autonomic nervous system [10]. These changes in pupil size are involuntary and automatic [3], and have been demonstrated to vary due to implicit cognitive processes. These processes include cognitive load, cognitive arousal, attention [6, 10–15], object recognition [16], memory retrieval [17–20], and novelty [6, 21, 22].

While pupillometry has been used in cognitive psychology since the 1960s (reviewed in [3, 6]), its use in other disciplines has been infrequent. This is most likely due to the difficulty of isolating the small relative change in pupil size due to cognitive processes (~1 mm) from the changes of up to 9 mm elicited by physical light/luminance [7] present in many applied settings. Since pupil size changes due to cognitive processes are small, traditionally only targeted experimental paradigms have been able to dissociate the effects of cognitive processing and physical stimulus properties on pupil dilation. These experimental paradigms attempt to keep the change in luminance between screens minimal by using grey-scaled images, movies or words as their stimuli of choice [19, 20, 22, 23]. Thus, well-controlled experiments are essential to preclude substantial confounds.

The extent of consumers' brand familiarity depends on their experience with a brand (i.e., brand usage) and is an indicator of brand awareness—one dimension of brand knowledge that is the basis of consumer-based brand equity [24]. Brand awareness is frequently assessed with self-report measures (reviewed in [1]) of brand recall (i.e., actively retrieving the brand based on a product-category cue; [24]) and brand recognition (i.e., remembering having seen the brand before when being exposed to a perceptual cue representing the brand, such as brand name, logo, or packaging; [24]). However, self-reports and implicit indicators, such as physiological responses not governed by conscious control, sometimes diverge. While researchers posit a positive relation between brand familiarity and consumers' preferences in brand selection [25], consumers reported to be undecided, even though automatic mental associations signifying unconscious beliefs predicted their future choice [26]. Another study reported that physiological responses (i.e., EEG and galvanic skin response) indicated differential responses to two advertisements, whereas participants did not consciously perceive this difference [27]. These examples underline the importance of complementing self-reports with objective measures of cognitive processes.

Several studies have shown that pupil responses capture familiarity with sensory stimuli in a general sense. Greater pupil dilation in response to old (vs novel) stimuli is a well-documented pupil effect [18, 22, 28–31]. However, inconsistent findings have also emerged. For example, Beukema et al. [23] observed participants' pupil responses to visually presented familiar and unfamiliar objects. The participants were required to identify whether a white stimulus presented on a grey background resembled an object, a non-object, or a random array of dots. The results showed that unfamiliar objects (i.e., objects composed of an array of randomized dots without a specific shape) elicited greater pupil dilation than the familiar objects [23]. Contradicting Beukema and colleagues [23], other researchers reported greater pupil constriction in response to retrieving memories of novel, colorful images of natural

scenes [21]. Despite inconsistencies, these studies indicate that pupillometry is suited for investigations of familiarity.

These inconsistencies appear to be a consequence of largely varying experimental designs and stimuli. Particularly in the context of applied pupillometry, there is a scarcity of unbiased experimental paradigms that preclude confounds due to sudden changes in luminance or color between screens to provide more objective information on what consumers experience, think and feel [1]. Exceptions are the use of large scrambled blobs of a movie still [32] that precludes access to the content of the movie, but may nonetheless lead to a pupil response depending on the part of the mask participants gaze at [33], and an array of square checks used with monkeys [34]. To address this issue, this study focuses on brand familiarity—a widely studied and relevant construct in the consumer psychology literature that relies heavily on self-reports—to illustrate the efficacy of a paradigm for pupil size measurements designed to dissociate smaller cognitive effects from larger physically induced confounds on a single-participant level across various stimuli. Building on the existing notion of scrambling content, this paradigm adapts the pupil to a product image's overall luminance and colors without revealing the product itself by presenting a pixel-scrambled image prior to the presentation of the intact version of the same image. Due to its flexibility, this paradigm can accommodate a variety of visual stimuli and be adapted for passive viewing (as illustrated in the present study), choice, or decision-making tasks requiring a behavioral response. Its application in combination with robust statistics [35–37] has the potential to clarify cognitive processes underlying consumers' decisions.

Based on physiological considerations and previous findings, we hypothesize that familiar brands elicit a greater change in pupil size starting around 500 ms post-stimulus onset when compared to an immediately preceding baseline period involving the presentation of a scrambled image of similar average luminance and identical pixels. Participants may experience an increase in processing effort when viewing product images of unfamiliar novel brands, as they attempt to retrieve brand information from memory. Because pupils dilate under increased cognitive load [6], we expect that viewing unfamiliar brands leads to transient pupil dilation, although the size of this dilation relative to familiar (i.e., old) product images remains to be determined. A lack of previous research precludes the development of predictions regarding specific product category effects. We speculate, however, that there will be differences between product categories, as categories themselves are associated with varying levels of familiarity. For example, young adults may encounter and use certain product categories to a greater (e.g., food) or lesser (e.g., cleaning products) extent. This likely affects familiarity with brands within the category and, in turn, pupil responses. Lastly, we do not expect familiarity differences during the baseline period, since the pixel-scrambled version of the product image does not carry any information with respect to products or brand familiarity.

## Materials and methods

### Participants

This research comprises two studies. In the first study, 763 students (605 female, 154 male, four preferred not to disclose; $M_{age}$ = 22.33; $SD_{age}$ = 4.34, $range_{age}$ = 18–53) at Concordia University completed an online questionnaire and self-reported their brand familiarity with 551 Canadian and foreign products. Of these 763 participants, 551 participants were permanent residents, 60 participants were temporary residents, and 152 participants were neither a permanent nor temporary resident of Canada. All participants received course credit as compensation for their time. This study received approval by the Concordia University's Human Research Ethics Committee (UHREC certificate 30000632).

An independent group of 17 Concordia University students (12 female, four male, one preferred not to disclose; $M_{age}$ = 22.94; $SD_{age}$ = 4.13, range$_{age}$ = 18–38) took part in the second study (i.e., the pupillometry experiment). This precluded possible confounding effects due to prior exposure to the branded product images included in the pupillometry experiment. The large sample responding to the questionnaire ensured that brand familiarity results were largely representative of those encountered among participants of similar education, age and residency status who took part in the pupillometry experiment. Additionally, the following inclusion and exclusion criteria applied to participants in the pupillometry experiment: self-reported normal or corrected-to-normal vision, no history of neurological disorders, normal amount of caffeine intake before the time of participation. Unusually large caffeine intake over the 48 hours prior to participation led to exclusion from the experiment, as caffeine ingestion can enhance pupil dilation for up to 6 hours if one is not used to it [38, 39]. Participants who reported the use of prescription, over-the-counter, or recreational drugs were excluded as well, since these drugs can affect pupil responses [7, 40]. As compensation, participants received either course credit or $5.

Of the 17 participants in the pupillometry experiment, two were excluded due to incomplete eye tracking data (i.e., one due to equipment failure, one due to participant dropout). Hence, the final sample included 15 participants (10 female, four male, one preferred not to disclose; $M_{age}$ = 23.2; $SD_{age}$ = 4.33, range$_{age}$ = 18–38). Similar to the first sample, most participants reported to be permanent residents of Canada (13 participants), with two participants reporting that they were neither permanent nor temporary residents of Canada.

## Stimuli

As a precursor to the pupillometry experiment, we administered an online questionnaire to identify a set of branded product images that differed in familiarity, as perceived by young adult consumers living in Canada. The current lack of a valid, experimental familiarity manipulation applying to Canadian participants, in combination with increasing effects of globalization on brand knowledge, were important reasons for conducting this validation study. Participants evaluated a set of 551 colorful product images on white background (750 x 750 pixels) that was retrieved from popular online stores in Canada and Europe, including product images from four categories (i.e., beverages, food, personal care, cleaning).

Each image was rated on four validation scales related to brand familiarity ("I feel very familiar with brand [brand name]", anchored 1 = strongly disagree, 7 = strongly agree), brand knowledge ("I know the product(s) of brand [brand name]", anchored 1 = strongly disagree, 7 = strongly agree), brand experience ("I feel very experienced with brand [brand name]", anchored 1 = strongly disagree, 7 = strongly agree), and specific product usage ("I have used the displayed product of brand [brand name] in the past", anchored 1 = strongly disagree, 7 = strongly agree; [41]). The first three statements referred to the brand whereas the fourth statement specifically referred to the product displayed. Of the full set of images, each participant only rated an average of 92 images to avoid fatigue effects. Quotas ensured that the number of collected data points was similar across all rated images (127 responses per image on average). In addition to rating these images on the four validation scales, participants answered 15 questions related to their own travelling, consumerism, and purchasing decisions that were interspersed throughout the experiment.

Since the brand familiarity score was used to identify the set of 300 images included in the pupillometry experiment, we ascertained that this scale captured different elements of brand familiarity represented by the other three validation scales. We therefore correlated the brand familiarity scale with each of the three other scales, and found that all robust bend correlations

were similar and close to one ($r(549) = .97$, $r(549) = .99$, $r(549) = .93$ respectively), with knowledge about the products of the brand showing the largest correlation (Fig 1a). Likewise, the distribution of the median ratings for each validation scale were similar, with extreme values being most prevalent (i.e., 1 = strongly disagree, 7 = strongly agree; Fig 1a). Thus, brand familiarity ratings seemed to capture these different facets of consumers' brand familiarity, and were used representatively as an indicator of brand familiarity.

The pupillometry experiment included a subset of 300 images. This subset consisted of 150 familiar and 150 unfamiliar images across the four product categories (Table 1), as determined by their median score across all participants on the seven-point brand familiarity Likert scale. Any image with a median score between 3.5 and 4.5 was excluded to reduce ambiguity. Again, high correlations between scores on the brand familiarity and the other three scales (brand

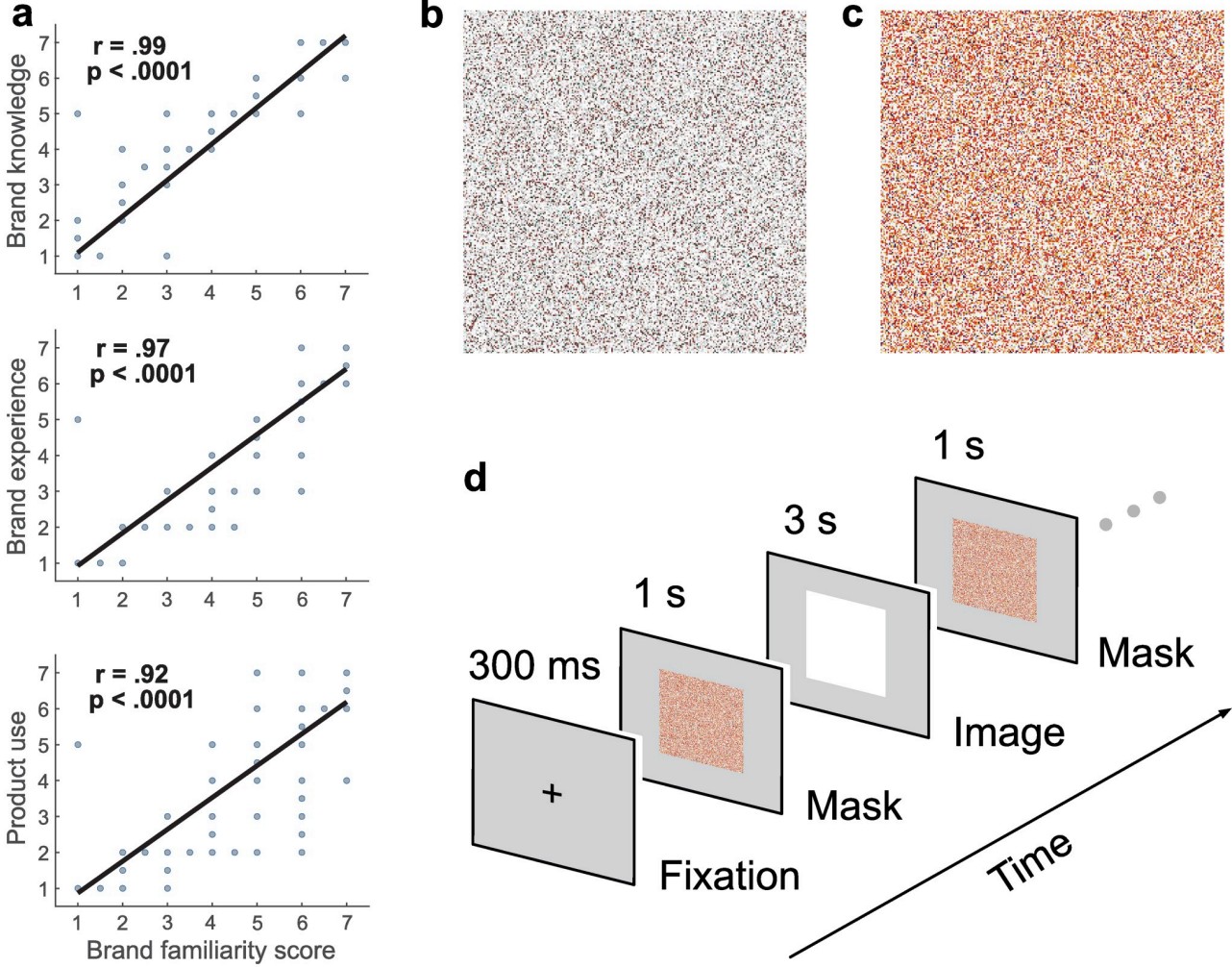

**Fig 1. Validation survey results, experimental paradigm and example of stimuli. a)** Median ratings for each of the three other validation scales (experience with the brand, knowledge of the brand's products, product use) correlated with the band familiarity ratings. Correlation coefficients and p values obtained from robust bend correlations; 5% downweighted [37]. Similar high correlation values show that image ratings were similar across scales. **b)** Example of a pixel-scrambled mask of a mostly white and brownish familiar branded product image (e.g., bag of brand coffee beans) included in the pupillometry experiment. **c)** Example of a pixel-scrambled mask of a colorful branded product image (e.g., container of chocolate powder). **d)** Sequence of events of one trial of the pupillometry experiment. Each trial began with a black fixation cross on a grey background. The intact image (i.e., familiar or unfamiliar branded image) was then framed by a pixel-scrambled version of the intact image. This product image is represented only by its white background but an intact product was presented on this background. Each trial lasted about five seconds.

**Table 1. Number of familiar and unfamiliar images selected for the pupillometry study.**

|  | Beverages | Food | Cleaning | Personal care |
|---|---|---|---|---|
| Familiar | 41 | 41 | 28 | 40 |
| Unfamiliar | 37 | 37 | 39 | 37 |

knowledge, brand experience, product use) for this subset illustrate the representative nature of the chosen scale ($r(298) = .996$, $r(298) = .988$, $r(298) = .972$, respectively; S1a–S1c Fig), After selection, an intact version as well as a pixel-scrambled version of the selected images were generated by a random pixel rearrangement via a custom MATLAB script (version 2020a, [42]). No stimuli were selected that are known to lead to changes in arousal [43].

## Procedure and paradigm

One sample completed the online validation questionnaire online. Upon selecting the subset of images of common consumer products, those participating in the pupillometry study were asked to attentively view these images on a computer screen without providing an overt response to each image. Participants were instructed to pay attention to the overall image, and were told that they would not have to search for or pay attention to specific elements. To ensure that participants engaged intentionally in the task, we randomly presented 10 attention trials that required a binary response—one at a time—throughout the session. These trials consisted of a simple question (e.g., "Do you prefer to shop online or in-store?"), and participants had to select one of two options by pressing either the spacebar or "0" on the keyboard. An eye tracker remotely collected participants' pupil responses throughout the entire session.

At the start of each session, a 9-point calibration was performed using the EyeLink's inbuilt calibration procedure with dots in default locations (version 4.56, SR Research, 2017, Ottawa, Ontario; [44]). Participants were asked to fixate on a series of nine equally spaced black dots (one degree in visual angle wide) on grey background. To validate calibration accuracy, the same nine points were presented in random order. Successful calibration required participants to achieve an average error of less than 0.5 degree of visual angle across all nine points, with no single point exceeding one degree of maximum error. Upon calibration, participants were presented with 150 familiar and 150 unfamiliar product images in randomized order. Each trial consisted of four phases. A black fixation cross on grey background (this background color likely avoids discomfort due to lower luminance levels and contrast compared to a white background [45]) was presented to participants in the centre of the screen (300 ms), followed by a scrambled version of the product image (1000 ms). Next, the intact version of the image was presented (3000 ms), followed by the same scrambled image (1000 ms). In total, each trial lasted 5300 ms (Fig 1d). The masks of scrambled images provided a measure of pupil response to physical image properties, including lighting and luminance, without the possibility of the semantic content of the intact object affecting the pupillary response. The pupil size change obtained in response to the intact image was therefore likely the result of a cognitive response to its content. The lighting was kept constant and participants' heads were stabilized on a chin rest throughout the experiment to preclude focal distance or head angle confounds. We avoided placing demands on working memory, as the task was designed to focus only on viewing the stimuli.

The experiment included block breaks every 60 images to mitigate potential effects of fatigue. These breaks were 45 seconds in duration. Participants were instructed to keep their head on the chin rest to avoid invalidating the initial calibration. The pupillometry study took about 30 minutes to complete, and thus reduced possible fatigue effects due to lengthy testing

sessions [7]. At the end of the session, participants completed a short online questionnaire about demographics, travel experience, and individual brand importance. For example, participants were asked "How do you usually choose a product while grocery shopping?". The purpose of these questions was to evaluate whether individual differences affect pupillary responses.

## Apparatus

An iMac (2011 27" i7, 16GB RAM) was used for stimulus presentation and data collection. Participants viewed the stimuli on a luminance and color calibrated video monitor (View Sonic, G225fb 21" CRT, 1024 x 768-pixel resolution, 100 Hz refresh rate). Participants were positioned 70 cm away from the monitor and rested their head on a chin rest to stabilize their head position. Participants' binocular eye position was collected remotely using a video-based eye movement monitor (EyeLink 1000, 1000 Hz sampling rate in binocular configuration if possible, running host software version 4.56, SR Research, 2017, Ottawa, Ontario; [44]).

## Data cleaning and baseline correction

To ensure that these pupil data stemmed exclusively from fixations on the product image and not the screen background, pupil data, collected at a sampling frequency of 1000 Hz, were obtained from an interest area set around the intact image including its white space (500 x 500 pixels) subtending ~4 x 4 degrees of visual angle.

Preprocessing of the baseline and product-viewing stage pupil data included the interpolation of blinks, data smoothing, subtractive baseline correction [46], excluding trials with numerous missing and/or outlier samples, and downsampling. Blinks were detected and interpolated using a two-step process. First, we used DataViewer's in-built algorithm for blink detection (version 4.1.1, SR Research, 2019, Ottawa, Ontario; [47]) and extended the identified blink period by 50 ms on either side before using a Piecewise Cubic Hermite Interpolating Polynomial (*pchip* function in MATLAB) to interpolate these missing data. Pchip interpolation is similar to cubic-spline interpolation, but provides a better fit for the present dataset. Second, remaining artifacts of rapid changes in pupil diameter, often due to small or partial blinks that did not lead to full tracking loss or other recording issues, were detected by a custom MATLAB script using the median absolute deviation of a moving window with a width of 50 ms that was shifted in increments of 30 ms [48]. These deviations were also interpolated using the pchip function. The slow nature of the pupil signal allows for interpolation without losing relevant information in these cases. In fact, changes in pupil diameter due to cognitive effects may start to show as late as 500 to 1000 ms post-stimulus onset [7].

Outliers were defined through a combination of visual inspection of the distribution of all de-blinked and baseline-corrected pupil size samples and its 95% confidence interval [46]. Our pre-processing resulted in the rejection of trials due to insufficient baseline data that would not allow for sensible interpolation of blinks (i.e., > 50% of the second 500 samples of the first mask missing; 3.13% of all 4500 trials), or more than 20% of all 3000 samples missing during the product-viewing stage [7] and/or manual rejection due to a null signal (2.53% of all trials). Lastly, trials with overly large pupil constriction compared to baseline were rejected as well (i.e., exceeding 3*SD below a participant's overall mean after baseline correction; 2.04% of all trials).

We performed subtractive baseline correction by subtracting the median value computed across the last 150 ms of the first scrambled image mask from all samples of the subsequent product-viewing stage [46]. This baseline period provided a measure of baseline autonomic tone under similar physical stimulus properties, which mitigates potential confounds due to

fluctuations of the autonomic nervous system [10] as well as those due to a change of the stimulus display's overall luminance and colors. The baseline-corrected 1000 Hz data was then downsampled to 30 Hz using a sliding window approach that has been used previously in the context of temporal EEG data [49–51] and suggested for pupillometry data [7]. Specifically, we computed the average pupil size for 50 ms windows (the default value implemented in other pupillometry analysis packages; [48]), with centers shifted in increments of 30 ms. This resulted in 99 windows representing the 3000 ms trial for further statistical analysis of a trial's mean and peak change in pupil dilation, and pupil size change over time proportional to one's pupil elasticity. In the present analysis, pupil elasticity refers to the difference between the minimum and maximum pupil size of a participant shown during the recording session, referred to as an individual's dynamic range in previous literature [7].

## Statistical analysis

The pupil responses to the branded product images were evaluated over time in terms of the change in pupil size during presentation of the intact image relative to the median pupil size during the last 150 ms of the first scrambled image mask. The statistical analyses entailed the computation of robust trimmed means (5% trimmed; [52]), percentile bootstrapped medians, effect sizes, and shift functions [35]. Trimmed means were calculated across trials of each experimental condition on the individual participant level. For the baseline period of each trial, after removing eye blinks, we computed one median value from 1000 Hz data that was then used during subtractive baseline correction. All statistics comparing measures of central tendency were complemented by robust effect sizes (i.e., Hedges' g, henceforth $g$) and their respective exact analytical 95% confidence interval, as computed by the *mes* function of the Measures of Effect Size Toolbox [36] in MATLAB.

To account for variation in baseline pupil size across participants and to prevent differences in individual pupil elasticity confounding group-level results, the pupil analysis entailed determining the dynamic range within each participant's trials of this study. In general, differences in dynamic range occur in populations differing in age [7]; however, to control for any potential confounds due to this factor, we chose the 2.5th and 97.5th percentile of the distribution of individual baseline-corrected minimum and maximum values as individual dynamic range. This range represented 100% of possible change around baseline [7]. All pupil figures and statistical comparisons were computed using baseline-corrected data showing the percentage change relative to baseline within an individual's dynamic range instead of using frequently reported arbitrary units. The latter are the default units when preprocessing pupil data with DataViewer (version. 4.1.1, SR Research, 2019, Ottawa, Ontario; [47]). Hence, we computed pupil size results on a single-participant level using baseline-corrected, downsampled (30 Hz) data before conducting statistical tests at the group level.

The main statistical analyses presented in Figs 2a and 3a, quantifying the difference between the response to familiar and unfamiliar intact product images, were conducted by means of a temporal cluster-based bootstrapping procedure. The bootstrapping techniques we use have been shown to be conservative, particularly with small sample sizes between 10 and 25 participants per group [53]. Our analysis used sampling with replacement of the individual familiarity condition difference scores (familiar–unfamiliar) of all 15 participants to generate one window-based, group-level median difference score resembling a score that could have originated by chance. Specifically, we obtained one median score across the difference scores of all participants for each of the 1000 bootstrap iterations of a sample/window. Another median was computed across all 1000 median scores of the latter distribution resulting in one group-level median difference score per window. In addition, the 95% confidence interval of the

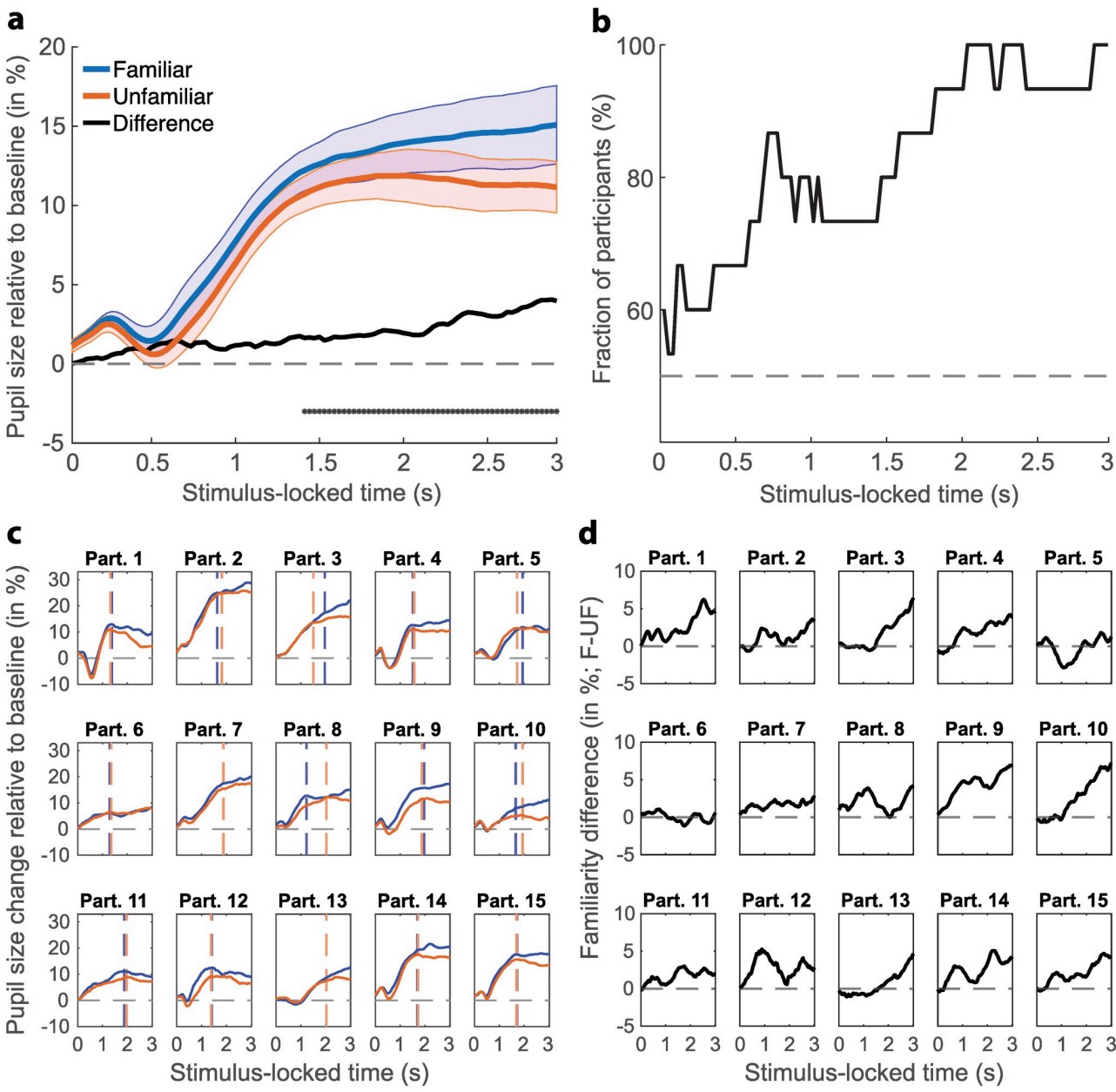

**Fig 2. Pupil size change relative to baseline and individual dynamic range. a)** Comparison of pupil responses to both familiar (blue) and unfamiliar (orange) intact branded images over time. Colorful lines show the 5% trimmed mean and shaded areas the bootstrapped standard errors of the mean. As a result of the temporal cluster-based bootstrapping procedure, the solid black line depicts the bootstrapped difference between the two familiarity conditions and black dots denote clusters of significant windows. The statistical test was conducted for every window with 30 ms onset intervals (i.e., 30 Hz). A data-driven minimum cluster size of 30 consecutive significant windows were required for a significant cluster. Dashed grey line depicts the median baseline pupil size of the last 150 ms leading up to the viewing of the intact image. The light blue and orange patches represent the bootstrapped standard errors for their respective familiarity condition. **b)** Fraction of participants showing an effect in the direction of the entire group; computed for each window. Dashed grey line depicts 50% of participant showing the same effect as the group. **c)** Individual participant-based comparison of pupil responses to both familiar (blue) and unfamiliar (orange) intact branded images over time. Vertical dashed lines indicate peak pupil responses within the first two seconds of viewing intact branded images by familiarity condition. If only one line is present both peaks were of identical latency. Color scheme as in panel a. All 15 participants shown. **d)** Individual difference between familiarity conditions (familiar–unfamiliar) relative to baseline and individual dynamic range in percent. Dashed grey lines depict the individual mean baseline across all trials of a participant. Note, black lines illustrate the effective difference and not the bootstrapped difference that is presented in panel a.

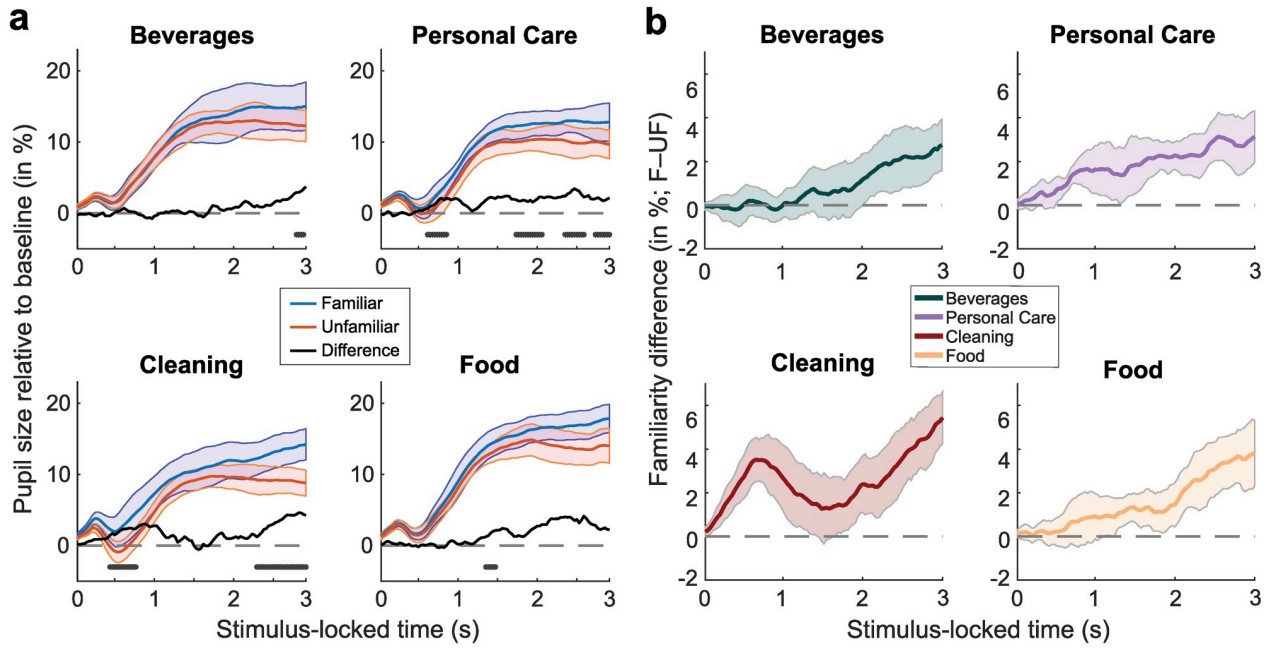

**Fig 3. Product category-based pupil size change relative to baseline and individual dynamic range. a)** Comparison of pupil responses to both familiar (blue) and unfamiliar (orange) intact branded images over time. Colorful lines show the group mean and shaded areas the bootstrapped standard errors of the mean for each condition. As a result of the temporal cluster-based bootstrapping procedure, the solid black line depicts the bootstrapped difference between the two familiarity conditions and black dots denote clusters of significant windows (for details, see Methods). Dashed grey line depicts the median baseline pupil size of the last 150 ms leading up to the viewing of the intact image. The light blue and orange patches represent the bootstrapped standard errors for their respective familiarity condition. **b)** Mean group difference between familiar and unfamiliar (familiar–unfamiliar) intact branded images in percent of the individual dynamic range for each of the four product categories: beverages (green), personal care (purple), cleaning (red), food (yellow). Shaded areas represent bootstrapped standard errors of the mean. Note, difference lines illustrate the effective difference and not the bootstrapped difference.

bootstrapped median difference scores was also taken from this distribution of 1000 median difference scores for each window. This confidence interval was then compared to zero, which is equivalent to a two-tailed test at an alpha level of 5% in the frequentist framework. If this 95% confidence interval did not include zero (representing zero or no difference) the window was determined significant. To correct for multiple comparisons, we repeated the above procedure 1000 times but randomly shuffled samples in time each iteration. We computed the size of the largest cluster of significant adjacent samples for each iteration and used the 95th percentile as minimum cluster size threshold. This data-driven procedure resulted in requiring a minimum of 30 consecutive significant windows (i.e., representing consecutive differences of ~950 ms of pupil data) as threshold for forming a significant cluster [54]. This threshold was lower for separate product categories (beverages = 3, personal care = 5, cleaning = 7, food = 5 windows). We performed this analysis using custom MATLAB scripts (version 2020a, [42]) that build on code associated with references [35, 55]. Since the temporal extent of significant effects using permutation analyses can only be approximately determined, we report a temporal range for the onset of significant differences in pupil size between familiar and unfamiliar product images [56].

Lastly, to obtain detailed quantitative information on the differences between two independent distributions, we used robust shift functions [35]. This technique compares each decile of the two distributions and estimates by how much and in which direction one distribution must be shifted to match the other one. This process provides insights beyond those offered by traditional point estimates, such as means or confidence intervals.

## Results

The final analysis of the pupillometry experiment included 2086 trials in the familiar condition and 2067 in the unfamiliar condition, resulting in an average of 521 and 517 trials per product category across all 15 participants. Results show a greater change in mean pupil size across the entire trial from baseline for familiar (vs unfamiliar) branded images ($t(14)$ = 6.03, $p < .0001$; $g$ = 0.44, 95% CI$_g$ = [0.230, 0.679]; $M_{Fam}$ = 9.90%, $M_{Unfam}$ = 8.11%; Fig 2a). This familiarity difference is further illustrated by one extended significantly different time period post onset of the intact image. A significant cluster emerged during the second half of the intact image viewing stage starting approximately between 1.3–1.5 seconds post-stimulus onset and lasted until the end (Fig 2a), although only approximate starting points can be given when using a permutation-based analysis approach [56]. This effect is represented by at least 75% of participants showing the same effect as the entire group (Fig 2b). This proportion increased with increasing viewing time illustrated by 90 to 100% of participants showing this effect consistently beyond 2 seconds post-image onset (Fig 2b).

Overall, these differences and their change throughout the viewing stage are corroborated by findings of larger first peak dilation in familiar trials ($t(14)$ = 5.44, $p < .0001$; $g$ = 0.36, 95% CI$_g$ = [0.179, 0.565]; $M_{Fam}$ = 13.96%, $M_{Unfam}$ = 12.13%; Fig 2c). Conversely, the results do not show a difference in first peak latency between responses to familiar and unfamiliar trials ($t(14)$ = -0.59, $p$ = .5647; $g$ = -0.16, 95% CI$_g$ = [-0.698, 0.381]; $M_{Fam}$ = 1676 ms, $M_{Unfam}$ = 1706 ms; Fig 2c). These results do not support the study's a priori hypothesis that unfamiliar brands elicit greater peak pupil dilation than familiar brands.

Of note is that we observed a qualitative, brief pupil constriction in the initial stages of the trial that was relatively small in magnitude (~2%; Fig 2a). This brief constriction follows the pattern of pupil size changes observed during object recognition [16] and may also be a result of the onset of the intact image with larger coherent white areas around the product. Overall, it did not differ significantly between both familiarity conditions, as determined by cluster-based permutation testing (Fig 2a).

It is conceivable that a potential familiarity difference present in the baseline values may have biased the observed difference during the viewing stage of the intact image. Therefore, we compared baseline values between familiarity conditions, but found no difference between these conditions ($t(14)$ = -0.62, $p$ = .5485; $g$ = -0.007, 95% CI$_g$ = [-0.030, 0.016]; $Median_{Diff}$ = 0.61, $SD_{Diff}$ = 12.7; in arbitrary units). Pupil size was virtually identical between familiarity conditions at the onset of the intact image ($M_{Fam}$ = 1.37%, $M_{Unfam}$ = 1.31%; Fig 2a). No significant difference was found between familiarity conditions for the median pupil size of a longer baseline of 500 ms either ($Median_{Diff}$ = 3.41, $SD_{Diff}$ = 12.64; $t(14)$ = -0.05, $p$ = .9596, $g$ = -0.0006, 95% CI$_g$ = [-0.024, 0.023]; in arbitrary units). Fixations with gaze angles that differ largely from the center of the screen could also result in pupil size differences due to optical distortion. However, we found the average deviation of gaze angles from the center of the screen to be virtually identical for both familiarity conditions ($M_{All}$ = 77.41, $M_{Fam}$ = 77.69, $M_{Unfam}$ = 77.12; $Median_{All}$ = 69.25, $Median_{Fam}$ = 70, $Median_{Unfam}$ = 68.64; in pixels). This result underlines that the observed differences are a result of the cognitive effects of differences in brand familiarity, as embodied by an intact and thus meaningful product image, and not potential differences in physical image properties between images distinct in familiarity.

We also examined the change in mean pupil size between familiarity conditions relative to baseline over time by product category. Product category is of importance in the field of consumer psychology since brand experience and exposure can affect consumers' brand recall based on their experience with the product category [57]. It is conceivable that product exposure and experience vary between the four product categories used herein. For example,

cleaning products might be used more infrequently compared to beverages. We found that pupil dilation shows a similar pattern across all product categories, with increasingly larger dilation in response to familiar products towards the end of the timeframe (Fig 3). Overall, these pupil size trajectories show a pattern similar to the general familiarity analysis. Significant temporal clusters were found for all product categories (beverages: around 2900 ms; personal care: around 700, 1800, 2500 and 2800 ms; cleaning: around 500–800 and 2300–3000 ms; food: around 1400 ms; Fig 3a). However, notwithstanding differences in the temporal locus of clusters indicating significant differences, we observed slightly different dynamics. While viewing beverages resulted in an increased pupil dilation difference in response to familiar vs unfamiliar products only towards the very end of the viewing period, viewing personal care, food, and cleaning products led to an additional increase in the pupil dilation difference before 1.5 s post-image onset. This earlier difference is the largest for cleaning products (Fig 3b) and appears to stem from a slightly increased initial pupil constriction in response to unfamiliar images of the cleaning category (~2%; Fig 3a).

As group-level results may hide differences present at the single-participant level, we evaluated the change in mean pupil size relative to baseline between familiarity conditions by product category at the individual level. Most participants exhibited a pupil dilation over time for most product categories (see S2 Fig). The change in mean pupil size relative to baseline varied by product category across participants nonetheless, and did not follow a consistent gradual pattern between categories (S2a Fig). Although these results illustrate that variability between participants' pupil responses exists within each category, the robustness of the direction of the effect was indicated by the high fraction of participants (up to 85%) who showed a difference similar to that of the group in all categories (S2b Fig). The varying patterns between product categories regarding the earlier difference—present for personal care and cleaning products— is corroborated by a larger fraction of participants (75–85%) showing the same difference during this viewing period (S2b Fig).

The change in mean pupil size relative to baseline aligns with the results from the validation questionnaire in that the more familiar product categories (i.e., beverages and food) show a similar trajectory, which differs from that of the more unfamiliar category (i.e., personal care and cleaning products). Particularly, the pupil dilation pattern in response to the cleaning products, showing a substantial difference early, aligns with the results from the validation questionnaire, as many of the cleaning products were categorized as unfamiliar (Table 1). This may have resulted in an amplification of the difference between familiar and unfamiliar images in this product category early on.

We also collected several scaled measures throughout the pupillometry experiment and in a questionnaire administered after the experiment. These related to consumerism, purchasing decisions, travelling, and relevance of brands for self-image, and were partly used for additional analyses. As the sample size was limited, only responses to the most relevant questions were analyzed using a bootstrapping approach to extrapolate these collected data. First, we observed that eight participants reported making product choices predominantly based on cost, and seven indicated that their product choice was driven by the brand name on this binary question. We split the sample based on their preference for selecting an item by cost or brand name. This group split was then used to compare participants' extrapolated maximum difference in pupil response to familiar and unfamiliar product images (i.e., familiar–unfamiliar, depicted in Fig 2d) in the form of the mean, median and 95% confidence interval of each subgroups' bootstrapped distribution of means (i.e., sampling with replacement to compute 1000 subgroup means; Fig 4a). This analysis revealed that participants who usually select their products by cost exhibited higher maximum pupil size difference between familiarity

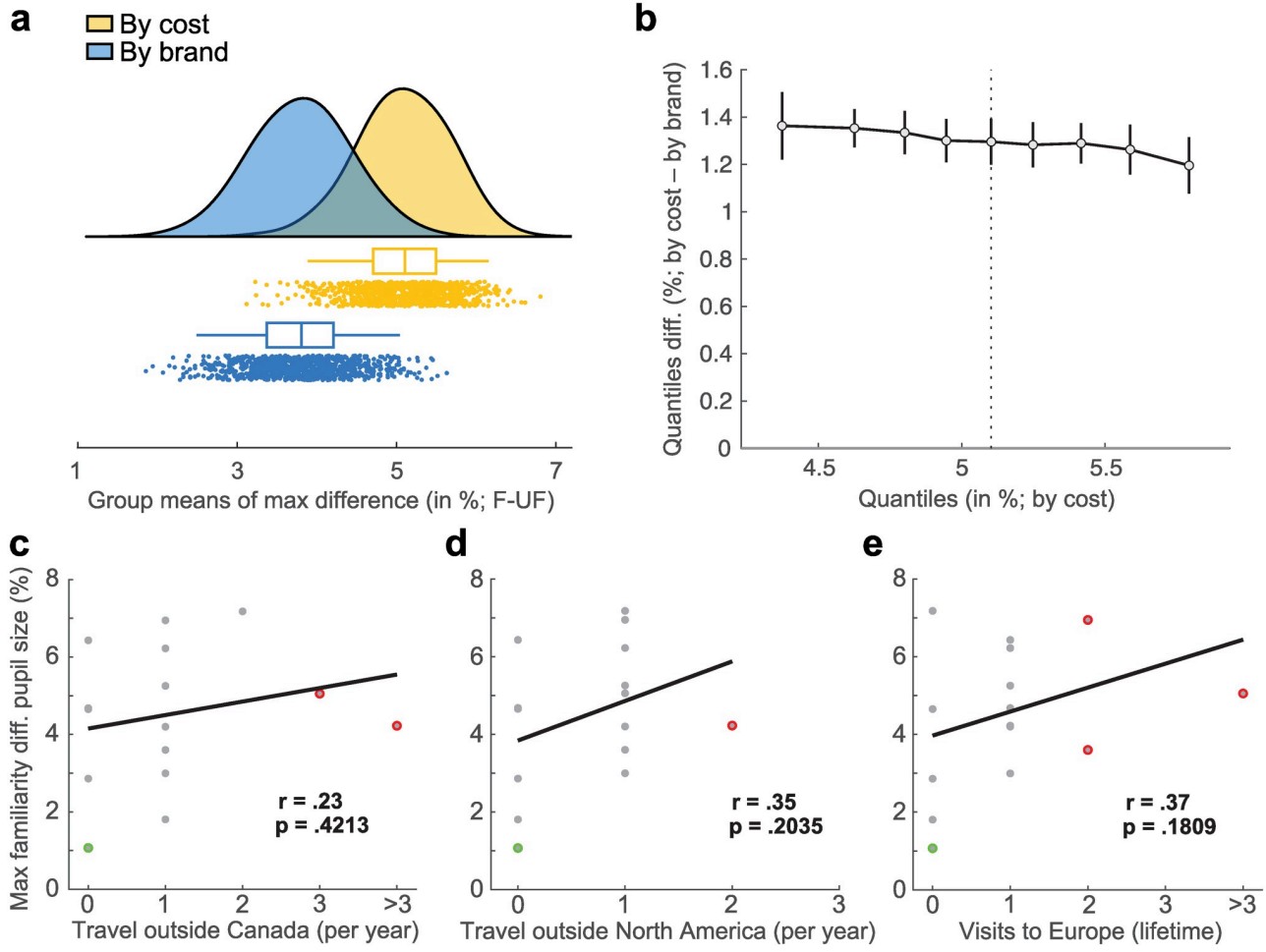

**Fig 4. Correlations between pupil data and attention questions. a)** Raincloud plot comparing bootstrapped means of subgroups split by self-reported usual product choice preference. Means generated from 1000 bootstrap iterations. **b)** Robust shift function comparing deciles of the two distributions shown in panel a [35]. The y-axis depicts by how much the "by brand" distribution (blue) has to be shifted along the x-axis to match the "by cost" distribution (yellow). **c—e)** Bend correlations between individual max pupil dilation difference and self-reported travelling frequency within different global contexts. Colored dots represent downweighted values in the x (red) and y direction (green) respectively. Some single-participant dots may be masked by dots of similar values.

conditions (Fig 4a). This difference was present and comparable in magnitude across all deciles when comparing both distributions (Fig 4b).

Moreover, we analyzed the potential influence of consumer ethnocentrism [58] and frequency of travel. Here, we observed a clear preference for buying Canadian brands, as theory would suggest ($M = 3.73$, median = 4, range = 1–5; [58]). In relation to travel frequency, at the time of data collection most participants reported travelling outside of Canada and North America once a year, and had been to Europe once in their lives (7, 8, 7 participants, respectively; Fig 4c–4e). Travel frequency did not correlate significantly with peak pupil size familiarity difference, as indicated by bootstrapped robust bend correlations ($r(13) = .23$, $p = .4213$, 95% CI [-0.270, 0.607]; $r(13) = .35$, $p = .2035$, 95% CI [-0.199, 0.726]; $r(13) = .37$, $p = .1809$, 95% CI [-0.241, 0.804], respectively; 5% of values downweighted; Fig 4c–4e). Lastly, participants reported that brands were of low relevance for their self-image ($M = 2.07$, median = 2, range = 1–4; options: not relevant | low relevance | medium relevance | high relevance). Taken

together, these results demonstrate that pupil size measurements are a robust indicator of individual-level differences in brand familiarity between domestic and foreign products across product categories, while not being significantly correlated with travel frequency to foreign countries.

## Discussion

The present study investigated cognitive familiarity differences via involuntary pupil size changes. Specifically, we examined pupil responses during the passive viewing of validated familiar and unfamiliar product images of common grocery store products. Results showed that familiar (vs unfamiliar) branded product images elicited a greater change in mean and peak pupil size. We identified one late temporal cluster that showed reliable differences using cluster-based bootstrapping analysis. These findings demonstrate that the employed pupillometry paradigm captures cognitive effects associated with brand familiarity robustly on the single-participant level, while ruling out confounds, such as the pupillary light reflex and effects of differences in image colors.

Pupillometry studies have traditionally avoided the presentation of colorful images, as changes in luminance and colors can result in large changes in pupil size [6]. These effects are up to 9 times larger than cognitively mediated pupil size changes. The paradigm used here ensures that pupil responses are a result of underlying cognitive processes rather than changes in physical stimulus properties. This is achieved by incorporating a pixel-scrambled version of the intact image prior to its presentation, which allows participants' pupils to adjust to the average physical image properties (i.e., luminance and colors) during this baseline period. This design allows for the inclusion of carefully selected colorful images in pupillometry studies. The application of this paradigm to the context of branded product images is particularly relevant, as these images are often colorful, and color carries crucial information for consumers. Research in this domain generally relies on self-report measures of consumers' brand familiarity, a method that is known to be subject to a range of biases [1]. Hence, measuring the cognitive effects of brand familiarity using involuntary pupil responses within the presented paradigm provides flexible opportunities for measuring pupil size in response to a variety of visual stimuli, colorful or not, in cognitive and consumer psychology.

Previous studies investigating pupil responses as a function of familiarity, the old/new effect, and object recognition have predominantly used grey-scaled images, movies or words [16, 19, 20, 22, 23, 59]—with few utilizing sounds [60, 61]. Two other studies have used largely varying colorful real-world scenes [21]. Generally, larger pupil dilation in response to old compared to new stimuli has been established across a variety of tasks [18, 22, 28–31]. It has been proposed that differences in familiarity underlie this effect [22, 23, 29, 60]. However, there are important distinctions in studies investigating the old/new effect and familiarity. These involve 1) whether familiarity or recollection is probed by a paradigm, and 2) whether active recall of information or retrieval of a feeling based on memorized information is required. Familiarity and recollection are two forms of memory that can lead to the recognition of a stimulus [29]. While recollection requires the retrieval of contextual information present at the time of encoding, familiarity represents the feeling of a memory of the stimulus without specific information linked to the time of encoding. Both types of memory support an old/new response in recognition memory tasks [62] and can be dissociated using pupil responses [20, 22].

To ensure that the observed effects were due to implicit differences in familiarity-related memories of the previously encountered products and not requiring recollection of contextual information, we opted for presenting product images without explicit task demands. Participants were instructed to attentively, but passively, view images and were not made aware of

the familiarity manipulation before the experiment. This precluded confounding effects due to varying levels of cognitive effort involved in recollection and active recall of where the product or brand had been previously encountered. This approach also circumvented the allocation of attention to specific aspects of the presented product images other than the information the consumer is naturally drawn towards. Hence, we consider the observed larger increase in pupil dilation during passive viewing of familiar product images, in line with the well-documented old/new response [31] and successful object recognition [16, 18, 20, 22, 63], a key advantage of and support for the sensitivity of the presented paradigm. If successful recognition of a familiar product occurred in this task, it was implicit and did not require the recall of contextual information.

The novelty of unfamiliar products is another conceivable explanation for the results, as novelty and surprise have also been found to elicit changes in pupil size [6, 20, 21]. In this study, novelty of an unfamiliar brand led to overall pupil dilation as well. This dilation, however, was considerably smaller in magnitude/amplitude compared to the dilation observed for familiar brands. This pattern of results is consistent with research showing that pupil dilation during the retrieval of novel items was smaller than during the retrieval of familiar items [63]. Naber and colleagues [21] also found that images of novel scenes (e.g., colorful houses, general landscapes) elicited weaker pupil dilation during the later phases of the retrieval period than images of familiar scenes. It has been proposed that this pattern of results arises because familiarity-based recognition is cognitively more effortful than an attempt to discern novel stimuli, also suggesting that encoding can form stronger associations with a given item [20]. Thus, while novelty of a stimulus can lead to pupil dilation, the paradigm presented here successfully replicated the old/new effect using colorful images by showing that pupil dilation to familiar (i.e., old) items is significantly larger than to unfamiliar (i.e., novel) items, without requiring an overt response. This pattern of pupil size changes also points toward deeper encoding of familiar items through prior exposure and experience. Greater exposure and experience with the familiar brands give rise to stronger brand associations and stronger brand familiarity [24], which were likely activated in our viewing task.

Another aspect worth discussing is that the employed bootstrapping approach identified temporal clusters showing a difference in pupil size between familiarity conditions that varied in their timing across product categories. Specifically, the difference in pupil size in response to cleaning products seems to be considerably larger early on during the viewing period, as illustrated by an early cluster around 500–800 ms post-stimulus onset showing significant differences (Fig 3a). This difference seems to be driven by a larger early pupil constriction viewing unfamiliar images of cleaning products. In comparison, although the later difference in pupil dilation that we observed across product categories is visible for all four product categories, differences in the variance of pupil size responses between categories may preclude statistical significance of a consistent large effect during the second half of the viewing period for beverages and food products on the individual category level (Fig 3a).

Evidence of temporal differences in the recognition of slowly emerging grey-scaled objects without explicit memory demands can provide useful insights [16]. This research found larger pupil dilation for recognized objects between 600 and 1500 ms (end of their viewing period) post-stimulus onset. This timeframe is in line with the increasingly larger dilation in response to familiar images towards the end of the timeframe that we observed across and within product categories after the initial small constriction. Firstly, the congruence of results suggests that our findings could reflect increased recognition of familiar compared to unfamiliar branded images, whereby participants indirectly confirmed that they had seen familiar brands before. Secondly, as consumers are better able to recall a given brand if they have personally used it [57], it is conceivable that the pupil responses to individual product categories may have been

affected by the usage frequency and depth of encoding of the products in each category. It is plausible that cleaning products were overall less familiar to the participants, despite these brands having been rated familiar in the validation questionnaire, as university students may have less or infrequent experience with these products or pay less attention to these brands when shopping. Consequently, cleaning brands may be less familiar (or more novel) on the familiarity continuum. This interpretation receives support from findings of larger pupil constriction when viewing subjectively novel scenes [21].

More evidence for the successful tracking of brand familiarity via pupil responses in the present paradigm arises from the stimulus context that underlies the robust familiarity effect observed for up to 100% of participants during individual temporal windows. Specifically, a certain baseline level of familiarity with the shape of product packages within the stimulus set is likely—even if the products and brands had never been seen before. For example, the shape of a beverage container offered in Canada and Europe is relatively similar or identical. Simply recognizing a product as a beverage should therefore have not differentiated familiar from unfamiliar brands. We purposefully decided against including unidentifiable objects, as other authors have done previously [23], to keep the levels of cognitive effort related to object identification itself low and comparable across trials, and to focus on the familiarity manipulation. Similarly, the absence of a difference in baseline pupil size between familiarity conditions for two baselines varying in length corroborates the notion that other familiarity-independent cognitive factors such as differences in attention, arousal or the images' scrambled version do not account for the observed difference in pupil responses. Thus, this study demonstrates that the presented, applied paradigm is capable of dissociating a cognitive effect (i.e., brand familiarity) from other commonly introduced stimulus- and task-related confounds.

On a neural level, the pupil responses to the branded images may reflect activity of the locus coeruleus (LC), its associated connections to several brain regions, and the Adaptive Gain Theory [64]. The LC is a subcortical structure that is linked to the brain's noradrenergic system and is found along the pupil dilation pathway. Cognitive processes that affect pupil dilation through the LC pathway likely involve contributions from other parts of the cortex as well (e.g., anterior cingulate cortex and the orbitofrontal cortex; [10]). These two structures are associated with the concept of utility [64]. The Adaptive Gain Theory suggests that the locus coeruleus-norepinephrine (LC-NE) modulatory system optimizes cognitive processes by adaptively conducting continuous evaluations of the targeted stimuli through the tonic (sustained) and phasic (event-related) modes of the LC. The phasic mode involves high task engagement and is activated in response to task-relevant events. In comparison, the tonic mode involves high distractibility and is associated with exploration [64]. In the current study, the results showed a greater increase in mean pupil size from baseline for familiar (vs unfamiliar) branded images while focusing on the task. This suggests that participants' LC neurons were in phasic mode when viewing the familiar branded images, as participants might have recognized relevant content. In comparison, participants' LC neurons may have been in tonic mode when presented with unfamiliar branded images, as the lack of familiarity may have led to greater exploration and distraction. The input signals from the anterior cingulate and the orbitofrontal cortex can generate increased activity in the phasic mode of the LC. Thus, it can be inferred that recognition of the familiar images may reflect an increase in the phasic mode of the LC, as familiar brands are thought to be more relevant to consumers [65] and may indicate higher utility.

The current study is not without limitations. Both the online validation questionnaire and the pupillometry experiment relied on undergraduate student samples. The nature of the sample is thus not representative of the general population. Although their prior experience with and exposure to the presented brands may differ compared to individuals from another age

cohort, sampling from a Canadian undergraduate population ensured that the results of the pretest applied to the independent sample that was selected for the pupillometry experiment. Moreover, the pupillometry experiment involved a rather small sample of participants (i.e., 15 participants). Although this is a potential limitation, the use of a large number of trials in combination with a careful choice of the statistical methodology (i.e., bootstrapping/sampling with replacement), and the robustness of the results in single participants (i.e., 14/15 participants showing a clear familiarity effect), suggest that the observed effects are likely to be replicable in a larger sample. In defining the familiarity conditions, we relied on the scores from the validation questionnaire, but did not ask participants in the pupillometry study for familiarity ratings, as these could have been influenced by stimulus presentation. However, the large sample responding to the online validation questionnaire (i.e., 763 participants) alleviates concerns regarding subjectivity of ratings, and the smaller sample participating in the pupillometry experiment. Another potential concern is that pupil dilation could have been a consequence of an illusion of brightness [66, 67] for certain images with salient bright spots. Since such bright spots were only present in some images, varied in size and location between images, and were not necessarily located towards the fixated centre of the product image, we consider this effect unlikely to be the cause underlying the observed pupil dilation effect. Lastly, another option for constructing the masks of unidentifiable images would be the use of phase scrambling in the Fourier domain to disrupt the structure and higher order image statistics. This technique is frequently implemented in face [68] and scene processing [69] and has its strength in rendering objects in grey-scaled images unrecognizable. We consider phase scrambling complimentary to the presented pixel-scrambled mask, as its effectiveness depends on the spatial frequency content of an image. At lower spatial frequencies phase scrambling tends to preserve the generic structure of images and can be classified based on its amplitude spectrum [69]. This could lead to larger preserved lighter and darker patches within the image. Given that product packaging often contains high amplitudes of horizontal and vertical information at low spatial frequencies, it may not provide the ideal solution for adapting the pupil without revealing any structural image content, which could act as a clue to a product's identity. Further, when phase scrambling is applied to separate RGB channels, it can also result in unrecognizable objects albeit larger areas of homogenous color (e.g., bright spots). If fixated, these brighter areas might counteract the goal of adapting the pupil to the overall luminance of the image during the baseline phase. However, since, to the best of our knowledge, phase scrambling has not been used in applied marketing research, this conjecture remains to be tested empirically in future studies.

Based on a validation study including familiarity self-reports and an eye-tracking study to obtain pupil responses, this research provides initial evidence for the validity of the pupil response as a measure of brand familiarity. It contributes to consumer research in several ways. First, despite the call to supplement traditional self-report questionnaires with other measures to better capture non-conscious processes and behaviors, particularly in the domain of visual processing [70], the use of such measures is still scarce. The current research demonstrates the value of such an approach and provides an initial multi-method triangulation of brand familiarity measures. Given that pupillometry measures can be obtained at the same time as other eye-tracking metrics, such as fixations and saccades, this approach is also promising in terms of its efficacy in a testing environment. Second, despite the slowly increasing use of eye-tracking in consumer research—mainly in obtaining processing measures of attention—pupillometry is rarely applied, despite the recognition of its potential value in measuring consumer responses, particularly in the context of advertisements [71, 72]. This may be due to a lack of an easily applicable measurement paradigm, which the present research sought to develop and test.

Prior to application of this paradigm in applied marketing research, it would be useful to measure both self-reported brand familiarity, single-trial behavioral and pupillary responses to unfamiliar and familiar brands at the individual consumer level to demonstrate that pupil responses successfully dissociate unfamiliar and familiar brands at the individual consumer and trial level. This approach would allow for the calculation of area under the curve measures to more clearly establish the sensitivity and specificity of the pupillary response as a measure of brand familiarity. In addition, a company may also use the response to a generally familiar product of the brand as a baseline for familiar stimuli. By applying the described approach, subsequent studies could then rely increasingly on pupillary responses to assess brand familiarity. Whether and how reliably this will be possible on the single-trial level remains to be seen. For the near future, a combination of pupillary responses with only a few carefully selected self-report measures seems to be most promising for consumer psychology research.

The value of employing pupillary measures in applied consumer psychology research lies in the increasing accessibility and affordability of eye-trackers, and technological developments (e.g., of remote eye-trackers). Increasing ease of eye-tracker use, along with the potential reduction of study duration and participant fatigue due to a reduction in self-report measures administered in testing sessions that also involve eye-tracking, enhances the appeal of pupillometry in applied research.

In summary, this study presents an adapted paradigm for investigating cognitive effects that circumvents common confounds introduced by stimulus properties, such as luminance and colors, by displaying a pixel-scrambled version of the stimulus prior to its presentation. This paradigm can be useful for investigating cognitive effects in response to various forms of static stimuli (e.g., black and white or colorful images) and different tasks (e.g., encoding phase, retrieval phase, tasks including a decision/response component or passive viewing). We demonstrate that the well-documented old/new (i.e., familiar/unfamiliar brands) pupil effect —represented by larger pupil dilation for old/familiar stimuli—can be detected during simple passive viewing of validated, colorful product images using pupillometry. This research further demonstrates the applicability of physiological measures to investigate brand-related memory, and shows that pupillometry without explicit task demands can identify differences in familiarity between branded product images. Our results suggest that pupillometry is a valid approach to the measurement of brand familiarity that circumvents the biases associated with consumers' self-reports [1]. Thus, the presented paradigm could be used to generate new insights into the cognitive processes underlying consumers' responses to brand information among other kinds.

## Supporting information

**S1 Fig. Correlation of validation scores of the experimental stimulus set. a-c)** Bend correlations of median scores on the brand familiarity scale with the brand knowledge, brand experience, and product use scales, respectively. Median scores of the 300 images that were used in the pupillometry study plotted. This is a subset of the median scores of all images obtained through the validation survey depicted in Fig 1a. Correlation coefficients and p values obtained from robust bend correlations [37]. Solid black lines depict best fit lines. Circles denote values downweighted in the x, y, and both dimensions (red, green, and black, respectively; 5% of values).
(PDF)

**S2 Fig. Pupil size change relative to baseline and individual dynamic range by product category. a)** Individual difference between familiarity conditions relative to baseline and individual dynamic range by product category in percent of an individual's dynamic range. Dashed

grey lines depict the individual mean baseline across all trials of this participant. Colorful solid lines illustrate the effective pupil size difference. Product categories are depicted as follows: Beverages (green), personal care (purple), cleaning (red), and food (orange). **b)** Fraction of participants showing an effect similar to the direction of the entire group for each product category. Results computed for each window. Dashed grey lines depict 50% of participant showing the same effect as the group. Color scheme as in panel a.
(PDF)

## Acknowledgments

We would like to thank Simrat Malhotra for their help with data collection.

## Author Contributions

**Conceptualization:** Léon Franzen, Bianca Grohmann, Aaron P. Johnson.

**Data curation:** Léon Franzen, Amanda Cabugao.

**Formal analysis:** Léon Franzen, Amanda Cabugao.

**Funding acquisition:** Léon Franzen, Bianca Grohmann, Aaron P. Johnson.

**Investigation:** Léon Franzen, Amanda Cabugao, Bianca Grohmann, Karine Elalouf.

**Methodology:** Léon Franzen, Amanda Cabugao, Bianca Grohmann, Aaron P. Johnson.

**Project administration:** Léon Franzen, Amanda Cabugao.

**Resources:** Léon Franzen, Amanda Cabugao, Bianca Grohmann, Aaron P. Johnson.

**Software:** Léon Franzen, Aaron P. Johnson.

**Supervision:** Léon Franzen, Bianca Grohmann, Aaron P. Johnson.

**Validation:** Léon Franzen, Amanda Cabugao, Bianca Grohmann, Aaron P. Johnson.

**Visualization:** Léon Franzen.

**Writing – original draft:** Léon Franzen, Amanda Cabugao, Bianca Grohmann, Aaron P. Johnson.

**Writing – review & editing:** Léon Franzen, Amanda Cabugao, Bianca Grohmann, Karine Elalouf, Aaron P. Johnson.

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
