## [Decision Letter · Decision Letter 0]

15 Mar 2021

PONE-D-21-05690

Individual pupil size changes as a robust indicator of cognitive familiarity differences

PLOS ONE

Dear Dr. Franzen,

Thank you for submitting your manuscript to PLOS ONE. After careful consideration, we feel that it has merit but does not fully meet PLOS ONE’s publication criteria as it currently stands. Therefore, we invite you to submit a revised version of the manuscript that addresses the points raised during the review process.

We look forward to receiving your revised manuscript.

Kind regards,

Manuel Spitschan

Academic Editor

PLOS ONE

Additional Editor Comments:

Please specifically address the reviewer's points on framing the research, limitations in the current phase scrambling techniques and the analytic strategy.

Journal Requirements:

2. We note that Figure 1 in your submission contain copyrighted images. All PLOS content is published under the Creative Commons Attribution License (CC BY 4.0), which means that the manuscript, images, and Supporting Information files will be freely available online, and any third party is permitted to access, download, copy, distribute, and use these materials in any way, even commercially, with proper attribution. For more information, see our copyright guidelines: http://journals.plos.org/plosone/s/licenses-and-copyright.

2.1.         You may seek permission from the original copyright holder of Figure 1 to publish the content specifically under the CC BY 4.0 license.

2.2.    If you are unable to obtain permission from the original copyright holder to publish these figures under the CC BY 4.0 license or if the copyright holder’s requirements are incompatible with the CC BY 4.0 license, please either i) remove the figure or ii) supply a replacement figure that complies with the CC BY 4.0 license. Please check copyright information on all replacement figures and update the figure caption with source information. If applicable, please specify in the figure caption text when a figure is similar but not identical to the original image and is therefore for illustrative purposes only.

Reviewers' comments:

Reviewer's Responses to Questions

**Comments to the Author**

1. Is the manuscript technically sound, and do the data support the conclusions?

Reviewer #1: Partly

Reviewer #2: Yes

2. Has the statistical analysis been performed appropriately and rigorously? 

Reviewer #1: Yes

Reviewer #2: Yes

3. Have the authors made all data underlying the findings in their manuscript fully available?

Reviewer #1: No

Reviewer #2: Yes

4. Is the manuscript presented in an intelligible fashion and written in standard English?

Reviewer #1: Yes

Reviewer #2: Yes

5. Review Comments to the Author

Reviewer #1: The manuscript describes an experiment into the effects of brand-familiarity on pupil size. The experimental paradigm was designed with care to ensure that the observed pupillometric effects could be attributed to brand-familiarity rather than some other aspect of the stimuli. Brand-familiarity was initially assessed in an independent online study with a large representative sample size (n=763), precluding the requirement to obtain self-report measures from participants in the main pupillometry experiment (n=15). The study found that pupil size was greater on average when viewing familiar vs. unfamiliar brands, and additional analyses revealed how this difference is modulated by brand category and consumer behaviour.

* Is the manuscript technically sound, and do the data support the conclusions?

For the most part, the study is technically sound, but I note the following issues:

1) The sample size for the pupillometry experiment is low, especially in light of the small effect size and claims of a ‘robust indicator of brand familiarity’ (lines 492—495). Consider validating this claim by collecting more data or, alternatively, providing further statistical evidence (e.g., from simulations) that the effect is not down to chance.

2) The abstract indicates that the primary intention of the paper is to introduce a novel paradigm for exploring brand familiarity, which relegates the actual research to an example application of the paradigm. Note also that previous pupillometry studies have used a scrambled image mask approach (e.g., Nuske et al., 2014).

3) There is limited discussion of how the observed effects may relate to established theory regarding the mechanisms of cognitive pupil control. For example, the role of the locus coeruleus, and Adaptive Gain Theory (Aston-Jones and Cohen, 2005) or ‘Network Reset’ (Bouret and Sara, 2005).

4) Provide further explanation for why it is interesting to explore the pupil data with respect to brand category. What are the practical advantages of knowing this?

5) Note that pupil size can be affected by illusions of brightness. Laeng et al. (2012) first showed this, but there have been replications with real-world images (e.g., Castelotti et al., 2020). The image of Ovomaltine in Figure 1c is actually redolent of the stimuli used by Laeng et al. (2012), and it is possible that brightness illusions could be affecting the data for individual stimuli. This is unlikely to pose a major issue for the study, but it is worth mentioning.

* Has the statistical analysis been performed appropriately and rigorously?

The analyses appear to be in good order, but I have the following recommendations:

1) In addition to Maris & Oostenwald (2007), cite Sassenhagen & Draschkow (2019) and follow their recommendations for the reporting of cluster-based permutation tests. It is my understanding that these tests indicate only whether an effect was, or was not, present in the data, and that they should not be used to infer the temporal locus of an effect. For example, lines 412-416 discuss the early vs. late significant clusters, but I would have thought it would be appropriate to report only the largest significant cluster. Also specify what software was used to perform the tests (e.g., custom implementation, MNE, Fieldtrip, etc.)

2) Clarify why only 2.5 s of data for the 3 s stimulus period were analysed and whether this affected the significance of the results.

3) It would be informative to see the pupil data for the scrambled image. Also, it would be helpful to comment on why the pupil is on average already 5% greater than baseline at the start of the stimulus period. Could this be effects of anticipation, with participants being able to accurately predict when the stimulus would appear? Note that a temporal jitter for the duration of the scrambled image may have ameliorated this affect by making it difficult for participants to predict the onset of the stimulus.

4) Pupil data with EyeLink systems are affected by optical distortion due to eye movements. It would therefore help to clarify how well participants were able to maintain central fixation, and to exclude trials with large eye movements. See Hayes & Petrov (2015) for more information on this.

Minor points:

1) Line 227 – It is inaccurate to say that ‘participants’ eyes’ were calibrated’, as it is the eye tracker that gets calibrated. Say rather that a 9-point calibration was performed.

2) Line 268 – report degrees of visual angle rather than pixels

* Have the authors made all data underlying the findings in their manuscript fully available?

Only the averages per participant are available on the OSF. Please make all of the underlying data available.

* Is the manuscript presented in an intelligible fashion and written in standard English?

Note the following minor issues:

1) Line 308: Clarify what is meant by ‘pupil elasticity’

2) Lines 54-56: consider revising sentence

3) Lines 236-237: elaborate on the claim that the background color likely avoids discomfort

4) Line 466: ‘bootrapped’

5) Lines 518-520: consider revising sentence

6) Lines 572-573: consider revising sentence

Reviewer #2: Major comments

This paper investigates whether familiarity with images of consumer products modulates pupil responses to the onset of these images. It is found that products or brands, that are unfamiliar to observers, evoke stronger pupil constrictions around 500-1000ms and/or relatively weaker dilation at later time points than familiar products. This finding is in line with previous findings but the design is novel because the authors tested the novel/old effect on pupil size in a more applied context (i.e., market research) and tried to circumvent distorting effects of image features. The paper is well written, the study design is clear and logic, and the results are relevant. However, some of the framing and claims need to be modified before I can recommend this paper for publication. First, I expect that readers would like to know how well familiar products can be dissociated from unfamiliar products by using the pupil as an objective measure. Second, the claims concerning the proposedly innovative character of the experimental design need to be toned down.

Major point 1.

Will market research truly benefit from pupillometry? I deem the applied context of this experiment as the most novel and remarkable aspect of the study, and, as such, wonder how well the pupil dissociates familiar from unfamiliar images on a trial level. I would appreciate it if the authors could provide e.g. area under the curve measures reflecting sensitivity and specificity of pupillometry as a signal detection method. My expectation is that it will be hard to decide per product whether it is truly familiar to an observer or consumer, making pupillometry a less applicable method for market research companies. Nonetheless, perhaps the authors can provide some guidelines on whether or not and how to apply pupillometry in practice.

Major point 2.

Currently the authors frame their paper mostly on the proposedly innovative aspect of their procedure, namely using a scrambled baseline image presentation before showing the target image. The idea is that pupil size already changes in response to the colors and luminance levels of the baseline image (see Gamlin et al., 1998, for pupil responses to colors and other features). When the target image appears thereafter, the pupil will only respond to the familiarity of the object, not to the change in feature content, which is here not of interest and adds unwanted variance to the pupil response and may potentially cover up the effect of familiarity. While I understand the argumentation, I see two problems with how this issue was addressed and “sold” to the readers. First, the scrambled images do, to some degree, but not fully control for luminance and other image features such as spatial frequency (Weiskrantz et al., 1998). For instance, brightness perception and its effects on pupil size highly depends on which regions of an image are gazed at (see Derksen et al., 2018, for subtle effects of brightness distributions across images). While the mean luminance (all pixels) of the scrambled baseline image may be equal to the luminance of the target image, the bright image borders are possibly less often fixated or covertly attended after the image is shown, still affecting the early pupil constriction and following dilation. Also, the currente scrambled images do not adapt the pupil to spatial frequency content and the presence of an initial pupil constriction around 500ms marks that some feature changes still evoke a pupil response (spatial frequency, contrast, foveal luminance, object contours, etc.). A phase-scrambled version of an object within the images would serve as better controls (e.g., phase-scrambled images are often used in face perception studies). Second, the use of baseline images is not innovative, but a standard procedure in perception and vision sciences. The authors should thus consider toning down their claims with respect to the innovative aspect of the procedure, crediting previous work on baseline control images, and discussing the limitations of their baseline implementation.

Minor comments

- 69-70 it will help the reader to clarify what is meant with implicit indicators. Do you mean measures from tasks like the implicit association task (IAT)?

- Please double-check the references (e.g. Sirois & Brisson misses the title).

References

Derksen, M., van Alphen, J., Schaap, S., Mathot, S., & Naber, M. (2018). Pupil mimicry is the result of brightness perception of the iris and pupil. Journal of cognition, 1(1).

Weiskrantz, L., Cowey, A., & Le Mare, C. (1998). Learning from the pupil: a spatial visual channel in the absence of V1 in monkey and human. Brain: a journal of neurology, 121(6), 1065-1072.

Gamlin, P. D., Zhang, H., Harlow, A., & Barbur, J. L. (1998). Pupil responses to stimulus color, structure and light flux increments in the rhesus monkey. Vision research, 38(21), 3353-3358.

Signed review

Marnix Naber

6. PLOS authors have the option to publish the peer review history of their article (what does this mean?). If published, this will include your full peer review and any attached files.

Reviewer #1: **Yes: **Dr Joel T. Martin

Reviewer #2: **Yes: **Marnix Naber

---

## [Author Response · Author response to Decision Letter 0]

21 May 2021

Thank you for giving us the opportunity to revise our manuscript. A detailed response can be found in the response document including specific action points. A brief summary of the most important points of this revision can be found below and in the revised cover letter.

In short, we have uploaded a revised version of our manuscript, a marked-up copy of this revised manuscript, and a detailed response to the reviewers including a number of confirmatory analyses. In this process, we have re-analyzed our data using a different baseline and revised all results and figures and provided new figure files. The main pupil size difference remained in this revised analysis. Importantly, we revised the framing of our paradigm, the language around all analytical results, and developed brief suggestions for future consumer psychology research using pupillometry. Phase scrambling as an alternative image manipulation technique is being discussed in the response as well as the revised manuscript.

---

## [Decision Letter · Decision Letter 1]

28 May 2021

PONE-D-21-05690R1

Individual pupil size changes as a robust indicator of cognitive familiarity differences

PLOS ONE

Dear Dr. Franzen,

Thank you for submitting your manuscript to PLOS ONE. After careful consideration, we feel that it has merit but does not fully meet PLOS ONE’s publication criteria as it currently stands. Therefore, we invite you to submit a revised version of the manuscript that addresses the points raised during the review process.

We look forward to receiving your revised manuscript.

Kind regards,

Manuel Spitschan

Academic Editor

PLOS ONE

Journal Requirements:

Additional Editor Comments (if provided):

I would kindly ask you to make the minor changes suggested by Reviewer 2.

Reviewers' comments:

Reviewer's Responses to Questions

**Comments to the Author**

1. If the authors have adequately addressed your comments raised in a previous round of review and you feel that this manuscript is now acceptable for publication, you may indicate that here to bypass the “Comments to the Author” section, enter your conflict of interest statement in the “Confidential to Editor” section, and submit your "Accept" recommendation.

Reviewer #1: All comments have been addressed

Reviewer #2: All comments have been addressed

2. Is the manuscript technically sound, and do the data support the conclusions?

Reviewer #1: Yes

Reviewer #2: Yes

3. Has the statistical analysis been performed appropriately and rigorously? 

Reviewer #1: Yes

Reviewer #2: Yes

4. Have the authors made all data underlying the findings in their manuscript fully available?

Reviewer #1: Yes

Reviewer #2: Yes

5. Is the manuscript presented in an intelligible fashion and written in standard English?

Reviewer #1: Yes

Reviewer #2: Yes

6. Review Comments to the Author

Reviewer #1: All comments and concerns were addressed. The manuscript is now in my opinion technically sound, with data that support the conclusions and rigorous statistical analyses. Underlying data have been made availalble and the manuscript is in standard english.

Reviewer #2: The authors have addressed all my comments. However, my main concerns regarding phase-scrambling as a better baseline and the question how feasible pupillometry is as an application in market reserach remain. Also, I don't think that the plotting of fixation points helps to check for a potential confound of gaze angle (larger gaze angles lead to smaller pupil sizes) and thus whether the pupil differences between the familiar and unfamiliar conditions are confounded by the radius of gaze angle (see point 4 of reviewer 1).

First, I do not agree with the authors argumentation why phase-scrambling is problematic. Oliva & Torralba's results refer to a computer vision algorithm trained to categorize natural scene images converted to the phase domain. This paradigm and task is totally different from humans categorizing phase-scrambled images, which is really really hard, if not impossible with 0% phase coherence. Zero percent phase coherence also gets rid of any clues about contour. That phase-scrambling is a relatively advanced technique, is also not a good reason not to choose it. I understand it is out of the scope of the study to redo the experiment with better baseline images. I only expected that the authors at least consider phase-scrambling as an improved baseline paradigm to be described in the discussion. Because the authors have removed any reference to their baseline method as novel and innovative, the current, rather simple baseline implementation isn't a major issue anymore.

Second, please correct me if I am wrong, but I understand from the new discriminant analysis that familiar versus unfamiliar images can be classified based on pupil size merely above chance performance (0.52>0.50). This low score basically suggests that pupillometry is not a feasible technique to determine a person's image familiarity and that it is much more efficient to just ask them how familiar observers are with the images. This should be discussed. Although this finding reduces the impact and novelty of the findings, I still appreciate the effort the authors have put in this experiment and paper, and because the paper generalizes previous findings to an applied setting, I think the paper remains relevant to the field of pupillometry and market research.

Third, I would like to respond to the reply to a point made by the other reviewer about how optical distortion due to the difference between gaze and eye-tracker angle on pupil size. I can imagine why the authors do not want to invest a lot of time in implementing a model and controlling for a potential confound, although personally I would try to at least get more insights into the potential presence of this confound, if not only to satisfy the reviewer and show appreciation for their feedback. I suggest that the authors calculate the mean gaze angle deviation from the center of the screen (i.e., average fixation radius; mean(sqrt((x_gaze-width_screen/2)^2 + y_gaze-height_screen/2)^2)) per familiarity condition and see whether this differs. As a second control analysis, the authors should plot pupil size as a function of horizontal and vertical gaze position. They can bin gaze positions across the screen locations and plot the average pupil size per bin. If any differences in pupil size are observed (between familiar and unfamiliar, and across fixated screen locations), I suggest to incorporate gaze radius, and/or horizontal and vertical gaze positions as nonlinear, covariates in the discriminant analysis to see how much it explains away the difference in pupil size between the conditions.

7. PLOS authors have the option to publish the peer review history of their article (what does this mean?). If published, this will include your full peer review and any attached files.

Reviewer #1: **Yes: **Joel T. Martin

Reviewer #2: **Yes: **Marnix Naber

---

## [Author Response · Author response to Decision Letter 1]

11 Dec 2021

First, as outlined in the decision letter, we have performed the control analysis requested by reviewer 2. These two control analyses of pupil size as a function of screen location clearly demonstrate that the location of the fixations did not affect pupil size in a considerable manner that is relevant for the results of the present study. That is, while average pupil size increases slightly with increasing distance from the center of the screen, only a few bins towards the edges of the screen show much higher average pupil size. Importantly, these bins only hold a few fixation samples across the entire study (< 20), and therefore, do not affect our results.

Second, we present our arguments about the use of phase scrambling as a baseline method, which we support by visual illustrations (please see the cover letter and response to reviewers documents). We argue that making a specific statement on which method is the "better" baseline method needs to be answered empirically by future research.

Overall, in light of the reviewers' comments, we have added brief specifications relating to the points above predominantly to the discussion section.

---

## [Editor Report · Decision Letter 2]

5 Jan 2022

Individual pupil size changes as a robust indicator of cognitive familiarity differences

PONE-D-21-05690R2

Dear Dr. Franzen,

We’re pleased to inform you that your manuscript has been judged scientifically suitable for publication and will be formally accepted for publication once it meets all outstanding technical requirements.

Kind regards,

Manuel Spitschan

Academic Editor

PLOS ONE
---

## [Editor Report · Acceptance letter]

11 Jan 2022

PONE-D-21-05690R2 

Individual pupil size changes as a robust indicator of cognitive familiarity differences 

Dear Dr. Franzen:

I'm pleased to inform you that your manuscript has been deemed suitable for publication in PLOS ONE. Congratulations! Your manuscript is now with our production department. 

Kind regards, 

on behalf of

Dr. Manuel Spitschan 

Academic Editor

PLOS ONE